# 30-day survival and injury severity in traffic accidents in Iran: A mixed-method study protocol

Amin Talebi[1], Hamidreza Khankeh[1,2]*, Mohammad Saatchi[1], Mehrdad Farrokhi[1], Timoor Hosseini[1]

1 Health in Emergency and Disaster Research Center, Social Health Research Institute, University of Social Welfare and Rehabilitation Sciences, Tehran, Iran, 2 QUEST Center for Responsible Research, Berlin Institute of Health at Charité, Berlin, Germany

* hrkhankeh@gmail.com

## Abstract

### Background

Today, one of the greatest global challenges, particularly in developing countries, is traffic accidents and their associated consequences. The severity of injuries from these accidents often leads to serious outcomes for victims, including long-term disabilities and high mortality rates. Deaths from traffic accidents may occur at the scene, during transportation, or in hospitals. In light of the existing challenges in coordination between law enforcement and healthcare systems regarding traffic accident fatalities in our country, this study aims to: 1. Determining the 30-day survival rate and its relationship with injury severity among traffic accident victims in Khorramabad in 2025. 2. Identifying key challenges and proposing effective solutions for improving traffic accident management in Khorramabad.

### Methodology

This mixed-methods study will be conducted in three phases: **1. Phase 1 (Quantitative):** A prospective cohort study will investigate the relationship between injury severity (as measured by scales such as ISS)and 30-day survival rates. **2. Phase 2 (Qualitative):** Purposive sampling will be used to conduct semi-structured interviews with stakeholders (victims, families, EMTs, healthcare providers, etc.). Qualitative data will be analyzed via inductive content analysis (Hsieh & Shannon approach) to identify challenges and solutions in traffic accident management. **3. Phase 3 (Integration):** Findings from Phases 1 and 2 will be finalized and presented to policymakers, managers, and experts. The Delphi method will be used to prioritize challenges/solutions and finalize evidence-based recommendations.Results will be disseminated through: Peer-reviewed publications in reputable journals. Policy briefs tailored for officials to improve traffic accident management.

**Data availability statement:** No datasets were generated or analysed during the current study. All relevant data from this study will be made available upon study completion.

**Funding:** The author(s) received no specific funding for this work.

**Competing interests:** The authors have declared that no competing interests exist.

## Background

Today, one of the most pressing global challenges, particularly in developing countries, is the prevalence of Road Traffic Accidents/Injuries(RTA/I) and their severe consequences [1]. These incidents pose a significant threat to public health, with mortality rates becoming a critical societal concern [2]. Traffic accidents impose a significant burden on healthcare systems, disrupt social and economic stability, and hinder national development [3]. Alarmingly, they are currently the eighth leading cause of death worldwide across all age groups and are projected to rise to the fifth leading cause by 2030 [4,5].

In response, the United Nations General Assembly has designated two consecutive decades (2011–2020 and 2021–2030) for global road safety initiatives aimed at reducing casualties, underscoring the urgency of addressing this issue [6]. The World Health Organization (WHO) emphasizes the need for immediate action to meet the 2030 target of halving road traffic deaths and injuries [7–9]. Road traffic injuries are the leading cause of death for individuals aged 5–29 years, with over half of fatalities occurring among vulnerable groups such as children, pedestrians, cyclists, and the elderly [7–9].

Beyond the human toll, traffic accidents incur substantial economic costs. In 2022 alone, global direct costs exceeded $518 billion. Additionally, as young and middle-aged individuals—key contributors to the workforce—are disproportionately affected, these accidents indirectly strain economic productivity, exacerbating societal and familial financial burdens [8,9].

Meanwhile, the Eastern Mediterranean region, with 5.5% of the world's vehicles, accounts for approximately 11% of global road traffic deaths, equivalent to about 345 people per day. Injuries from these accidents were among the top 10 causes of death in the region between 2000 and 2019. It is estimated that if not controlled, they could rank higher by 2030 [10].

Among Eastern Mediterranean countries, Iran has the highest per capita car ownership (one car for every four people) and the highest traffic fatality rate per 10,000 cars: 33 deaths in Iran compared to 3 deaths globally. Traffic accidents are the second leading cause of death in Iran, accounting for 44 deaths per 100,000 people (approximately 17.7% of all deaths) [8,9,11].

According to the latest 2023 report from Iran's Forensic Medicine Organization, deaths and injuries from traffic accidents increased by 5.6% and 4.4%, respectively, compared to 2022 [12]. Additionally, traffic accidents impose a significant economic burden, costing Iran 8.4% of its gross national product—more than double the global average of 3% [13,14].

Mortality in traffic accidents is influenced by a range of factors, including gender, age, seat belt usage, driver's airbag deployment, vehicle type/size/age, driver's license status, lighting/weather conditions, loss of vehicle control, road-related factors, fatigue, injury severity, and the quality of medical services [15–17]. Notably, studies in Iran highlight the role of substandard medical care as a contributor to traffic-related mortality [17]. To evaluate injury severity—a critical determinant of mortality—scoring systems, such as the Injury Severity Score (ISS) are widely

employed [18]. These tools not only aid in predicting survival outcomes and assessing the quality of trauma care but also enable standardized comparisons of patients with varying injury patterns [19]. Furthermore, analyzing and predicting survival through time-to-event endpoint studies (e.g., time until death or recovery) can inform targeted preventive measures and improve therapeutic interventions, thereby reducing mortality [20,21].

Traffic accidents pose a significant global challenge, prompting extensive research worldwide, including in Iran. While deaths from these accidents can occur at the scene, during transport, or in the hospital [22], the relationship between survival rates and injury severity at these critical stages—initial scene, transportation, and hospitalization—has been understudied. Existing research has focused on specific aspects of survival, such as: Outcomes for patients transported by Emergency Medical Services (EMS) [3]; The impact of prehospital time intervals on survival in severe trauma [23]; Survival rates linked to advanced prehospital care interventions [24,25]; The role of prehospital physician presence in severe trauma outcomes [26]; The influence of trauma systems on survival post-accident [27]; Survival trends in severe traffic collisions [21] which failed to resolve the conflicting views between the health system and the traffic police.This body of work highlights targeted insights into prehospital and systemic factors, yet a comprehensive analysis of survival dynamics relative to injury severity across all phases of care remains lacking. Therefore, it seems that the relationship between the severity of injury in traffic accidents and survival rates in our country has not been thoroughly investigated. Suppose this phenomenon is studied in depth, thereby gaining knowledge of the factors affect death and injury in traffic accidents. appropriate practical recommendations can be developed based on scientific evidence to improve conditions and provide effective services.

### Aims

The aims of this study are:

1. Determining the 30-day survival rate of hospitalized traffic accident victims in Khorramabad in 2025, examining its association with age, gender, type of road user, vehicle type, injury mechanism, and accident location.

2. Determining the 30-day survival rate and its relationship with the severity of injury at the scene of the accident among traffic accident victims in Khorramabad in 2025.

3. Determining the 30-day survival rate and its relationship with the severity of injury during transport by emergency medical services among traffic accident victims in Khorramabad in 2025.

4. Determining the 30-day survival rate and its relationship with the severity of injury during hospitalization among traffic accident victims in Khorramabad in 2025.

5. Determining and comparing the severity of injury among road (inter-city), urban, and rural traffic accident victims in Khorramabad in 2025 by accident location.

6. Identifying key challenges and proposing effective solutions for improving traffic accident management in Khorramabad.

7. Prioritizing identified challenges and solutions for traffic accident management in Khorramabad through expert consensus and providing evidence-based applied recommendations.

### Methods/design

This study adopts a **mixed-methods design** (quantitative and qualitative) combined with expert consensus, structured into three sequential phases (Fig 1):

This figure will illustrate the three-phase research design. Phase 1 will consist of a quantitative cohort study. Phase 2 will involve a qualitative content analysis to explore participants' perspectives. Phase 3 will employ the Delphi technique to achieve expert consensus based on the preceding phases.

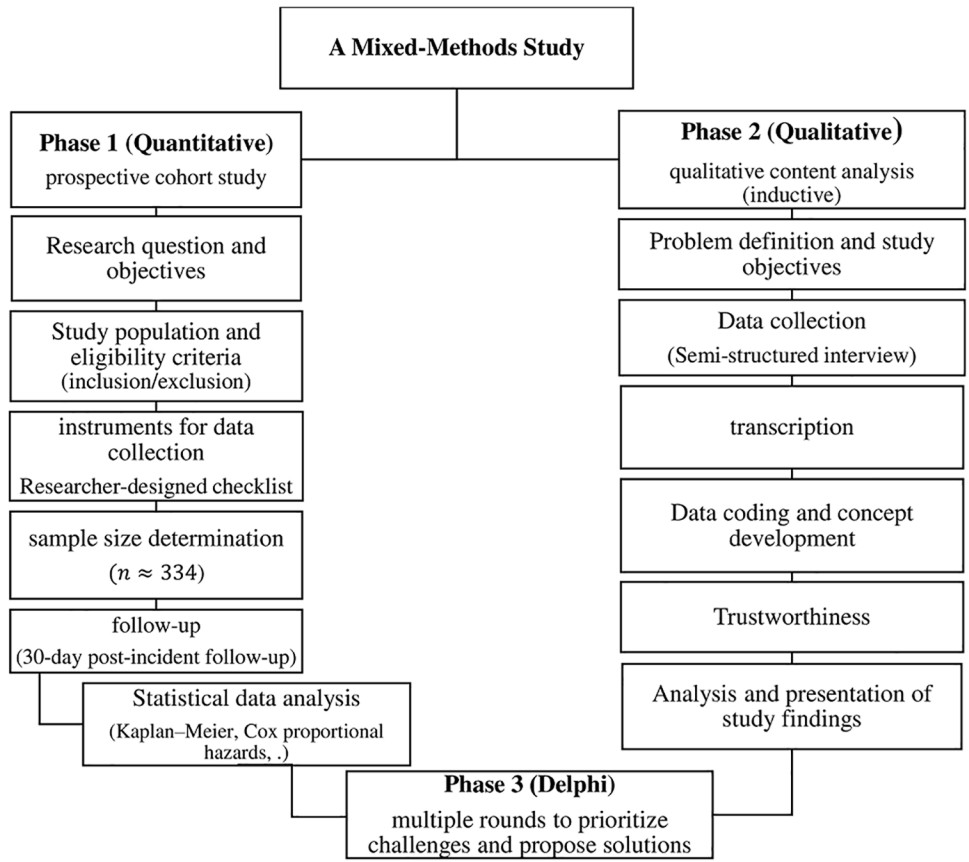

**Fig 1. Study steps and data collection.**

## Phase 1: Cohort Study

**Objective.** To investigate the 30-day survival rate and its association with injury severity among traffic accident victims admitted to Khorramabad in 2025.

**Methods. Study design.** Prospective cohort study.

**Setting.** Conducted at the Nomadic Martyrs Educational, Research, and Treatment Center (a level 1 trauma center per trauma center classification guidelines [28]).Includes traffic accident victims in Khorramabad City, Lorestan Province, transported by prehospital emergency medical services (EMS).

**Study population.** All patients with injuries from traffic accidents meeting inclusion criteria.

**Inclusion criteria.**

1. **Age ≥ 18 years**

Physiological responses to injury differ between adults and children; anatomically based Injury Severity Score (ISS) may overestimate mortality in pediatric populations [29].

2. **Transport via EMS**

Patients directly transported from the accident scene by EMS

**Exclusion criteria.** 1. Patients transferred from other medical facilities or non-EMS transport (e.g., private vehicles).

Sample size [30]

$$n = (Z\alpha/2 + Z\beta)^2 \ (p(1-p)) \ (\log(HR))^2$$

$$n = \frac{(1.96 + 0.84)^2 \quad \cdot (0.05(1-0.05))}{(\log(1.08))^2}$$

$$n \approx 334$$

**Data collection phase.** After the participant fills out the written consent form, the researcher will review the form for completeness and confirm that the participant understands the study procedures and their rights. Once confirmed, the participant will be officially enrolled in the study, and the research procedures will begin.

After obtaining approval from the Ethics Committee of the University of Social Welfare and Rehabilitation Sciences, receiving a letter of introduction from the university, and securing permission to conduct the research from Lorestan University of Medical Sciences, we will visit the 115 Emergency Center of Lorestan Province and the selected trauma hospital in Khorramabad. During these visits, we will provide the necessary explanations about the study and obtain final authorization to begin sampling.

In this study, the researcher will train emergency medical technicians on the Injury Severity Scoring System (ISS), including how to calculate ISS scores and how to complete the patient information checklist. Emergency medical technicians will fill out this checklist for patients at the scene of the accident, while the researcher will record the corresponding hospital information in the same checklist.

Finally, after 30 days, the researcher will follow up on patients who were transferred to the hospital. This follow-up will be conducted through a review of patient records and, if necessary, by contacting the patients directly.

The study will categorize patients into the following groups based on outcomes:

1. **Patients who died during the study period.**

2. **Censored patients** — those who were withdrawn or lost to follow-up. This group includes patients who:

   • Did not experience a relevant outcome (death or recovery) by the end of the study,

   • Were lost to follow-up,

   • Experienced an unrelated event preventing further follow-up, or

   • They Were discharged before 30 days and later died of another cause.

3. **Patients who survived and recovered within 30 days.**

In this phase of the study, data will be collected using a researcher-developed checklist designed to capture the following categories:

1. **Demographic information**: Contact number, age, gender, history of drug/alcohol use, medication history, user role (e.g., driver, passenger), and chronic disease history.

2. **Injury characteristics**

   • Accident Details: Vehicle type, injury mechanism (e.g., collision, fall), accident location/type, and injured body regions/limbs.

   • Clinical Metrics: Injury type (blunt vs. penetrating), arterial blood oxygen levels, vital signs, Glasgow Coma Scale (GCS) score, Brief Head Injury Score, and Injury Severity Score (ISS).

3. **Treatment measures**

- Prehospital Care: Initial interventions (e.g., airway management via jaw thrust for obstructions, breathing assessments, circulatory support) and transport method (ambulance, ambulance bus, or helicopter).

- Hospital Care: Treatments administered in the emergency department, hospital admission unit, mechanical ventilation use, length of hospitalization, and discharge/death outcomes (including time/date).

4. **Patient status tracking**

- Time/date of prehospital and hospital admission, duration of care, and final patient disposition (discharge or death).

**Injury severity assessment.** The Injury Severity Score (ISS) will be calculated and documented in the checklist to quantify the severity of trauma. This metric synthesizes data from multiple body regions to generate a standardized score, enabling consistent evaluation of the impact of injuries across various body regions.

**Statistical analysis.** Descriptive statistics will be used to summarize the demographic and clinical characteristics of the participants, including age, gender, mechanism of injury, injury severity score (ISS), type of injuries, and time to hospital. Depending on the distribution of the data (normal or non-normal), quantitative variables will be presented as mean±standard deviation or as median and interquartile range (IQR). Qualitative variables will be reported as counts and percentages.

Kaplan-Meier survival analysis will be used to estimate 30-day survival. The survival function will be plotted, and differences in survival between groups (e.g., different ISS categories) will be assessed using the log-rank test—Patients who survive beyond 30 days or are lost to follow-up before this period will be censored.

A Cox proportional hazards regression model will be employed to examine the association between injury severity and 30-day mortality. Initially, a univariate Cox analysis will be conducted to assess the independent effect of ISS on survival. Variables with a p-value < 0.2 in the univariate analysis, along with clinically important factors such as age, sex, comorbidities, and time to final care, will be included in a multivariate Cox regression model to adjust for potential confounders. In Cox regression, in addition to examining the relationship between individual variables and potential confounders, we also assess the interaction terms of key variables such as age, gender, and injury severity. The proportional hazards assumption will be tested using Schoenfeld residuals, and time-dependent variables will be included when appropriate. Results will be reported as hazard ratios (HRs) with 95% confidence intervals.

Subgroup analyses will be conducted to explore relationships within specific populations, such as different age groups, genders, or types of injury. Sensitivity analyses will also be performed to assess the robustness of the findings and to evaluate the impact of missing data and potential model assumptions violations.

**Ethical considerations.**

- Ethical approval (IR.USWR.REC.1404.037) was has been obtained from the University of Social Welfare and Rehabilitation Sciences (Tehran, Iran) on 28 May 2025.

- Patient confidentiality is ensured through the collection of anonymized data.

## Phase 2: Qualitative Study Component – Inductive/qualitative Content Analysis

**Objective.** To explore the challenges of individuals involved in traffic accidents, those responsible for managing traffic accidents, and stakeholders tasked with proposing corrective solutions based on their experiences and perceptions.

**Qualitative content analysis approach.** In this phase, qualitative content analysis with an inductive approach, utilizing the Hsieh and Shannon model, will be employed to thoroughly examine the experiences, understandings, and perspectives of experts and other stakeholders. The focus will be on identifying challenges and obstacles in traffic accident management and proposing corrective solutions [31].

Qualitative content analysis is a method of interpreting qualitative data by systematically summarizing, describing, and extracting meaning from textual material [32]. It is particularly suited to exploring participants' in-depth experiences and perceptions. While all aim to interpret textual meaning, the inductive approach is most appropriate when existing theoretical frameworks for a phenomenon are limited. This approach avoids imposing predetermined themes, allowing categories and subcategories to emerge organically from the data [33].

**Research community: Participant selection and sample characteristics.** In this study, participants will be selected purposefully, based on predefined criteria. After obtaining official permissions and explaining the study's objectives to eligible individuals, the researcher will conduct semi/structured interviews to gather data.

**Inclusion criteria.** Participants must meet the following conditions:

- **Relevant expertise or experience:**

  - Professionals in traffic accident-related fields, including:

    - Traffic and driving specialists

    - Academic researchers in traffic accident studies

    - Emergency medicine physicians, surgeons, and nurses

    - Emergency medical technicians (EMTs)

    - Police officers

    - Individuals present at accident scenes (e.g., witnesses, injured persons, patient companions)

- **Willingness to participate:** Demonstrated interest in contributing to the study.

- **Communication ability:** Capacity to articulate experiences, opinions, and views effectively.

**Exclusion criteria.** Participants will be excluded if they:

- Demonstrate an inability or unwillingness to continue cooperation during the research process.

**Participant selection method.**

- **Sampling Strategy:** Purposeful sampling with maximum diversity will be employed to select individuals with substantial knowledge or experience related to traffic accidents.

- **Sample Size:** The number of participants is not predetermined. Data collection will continue until saturation is achieved, i.e., until no new insights emerge from the interviews [34].

**Rationale for approach.** Semi-structured interviews will enable in-depth exploration of participants' perspectives, aligning with the qualitative focus on understanding lived experiences and expert insights.

**Data collection method.** After the participant completes the written consent form, the researcher checks that all required sections have been completed and that the participant's questions have been answered. This study will utilize Semi-structured interviews as the primary data collection method. The process is structured as follows:

**1. Interview design and protocol**

Prior to conducting interviews, the research team will develop a flexible interview guide containing core questions, probes, and follow-up prompts tailored to participants' responses. To ensure ethical compliance, the researcher will:

- Obtain written or verbal consent (with audio documentation) from participants.

- Clearly explain the study's purpose, confidentiality protocols, and participants' right to withdraw.

- Disclose approval from the University of Social Welfare and Rehabilitation Sciences and provide proof of research authorization upon request.

## 2. Participant selection and ethical considerations

Participants will be purposefully selected based on predefined inclusion criteria. Before each interview, participants will receive a standardized information form detailing:

- The study's objectives and ethical safeguards.
- Personal data to be collected (age, gender, education level, employment history, interview duration).
- Assurance of anonymity, secure data storage, and freedom to pause or reschedule interviews.

## 3. Interview procedure

Interviews will be conducted face-to-face in a private setting to ensure psychological safety and encourage open expression of experiences, thoughts, and perceptions. Key procedural steps include:

- Reiterating the interview's purpose at the outset.
- Allowing participants to set the pace; interviews may be paused or resumed later if fatigue arises.
- Concluding with an open-ended invitation for participants to share any additional reflections.
- Collecting contact information for follow-up if needed.

## 4. Data management and transcription

All interviews will be audio-recorded with permission. Immediately after each session, verbatim transcriptions will be produced. The researcher will:

- Review recordings and transcripts multiple times to identify nuances in participants' experiences.
- Supplement transcripts with observational field notes (e.g., nonverbal cues, contextual details).

## 5. Analysis

Qualitative content analysis, employing an inductive approach, will be used to analyze the qualitative data. In the first step, interview transcripts will be prepared verbatim and serve as the primary research data. In the second step, the texts will be divided into semantic units, which will then be summarized and condensed. In the third step, abstracting of semantic units and code selection will be conducted. Both explicit and implicit concepts, drawn from participants' statements, will be identified in sentences or paragraphs and assigned codes, followed by coding and summarization. In the fourth step, through continuous comparison of similarities, differences, and relationships, codes representing a single concept will be grouped into categories and subcategories. Ambiguous points will be clarified through follow-up interviews with participants, ensuring that all codes are accurately placed within the appropriate categories. In the fifth step, at the interpretive level, categories will be abstracted, central concepts identified, and the main and overarching themes extracted. These concepts will be reviewed to ensure that the internal themes accurately represent the entire dataset [33,35].

## 6. Evaluation of the accuracy and robustness of the study

In this study, we apply Guba and Lincoln's (1985) criteria to evaluate trustworthiness, which includes validity, reliability, confirmability, and transferability [36,37].

**Validity.** The researcher enhances the credibility of the findings through prolonged engagement, immersion in the data, and verification by both participants and observers.

**Reliability.** All interviews are recorded, transcribed verbatim, carefully documented, and reviewed by experts.

**Confirmability.** Every research step and decision is systematically documented to allow for transparency and future follow-up.

**Transferability.** Detailed descriptions of the participants, research context, and data collection procedures are provided to strengthen the generalizability of the findings.

## Phase 3: Delphi

**Objective.** Identify and prioritize challenges and obstacles to improving traffic accident management, and propose corrective solutions informed by expert opinions and knowledge.

**Methodology.** In this phase, three rounds of the Delphi technique are conducted to prioritize challenges and obstacles to enhancing traffic accident management and to develop corrective solutions derived from the outcomes of Phases 1 and 2. This approach leverages expert insights to ensure informed, consensus-driven results.

The Delphi technique is a structured process used to formulate predictions, aid decision-making, collect data through iterative survey rounds, and ultimately achieve group consensus. Unlike conventional surveys that focus on answering "what is," Delphi addresses "what could or should be," making it a robust method for generating actionable recommendations. Given the goal of identifying challenges and proposing solutions for traffic accident management, the Delphi method offers a cost-effective way to synthesize expert perspectives [38].

In practice, the technique involves sequential sessions with a panel of experts, incorporating structured feedback to foster consensus on a specific topic [39]. Controlled iterations ensure refinement of ideas while minimizing biases, aligning stakeholders toward evidence-based strategies.

## Conclusion

This protocol outlines the steps for conducting a mixed methods study to investigate the 30-day survival rate and its relationship with injury severity among traffic accident victims in Khorramabad in 2025, as well as to identify challenges and applied recommendations for managing them. Publishing research protocols prior to conducting a study can enhance methodological rigor and, according to researchers, strengthen the overall quality of research. The purpose of this study is to assist health system managers, policymakers, and organizations involved in traffic accident management in determining the causes of death among patients with traffic accident-related injuries, To raise awareness among the general public, students, and relevant organizations about traffic accidents, and to educate them on factors influencing injury severity, to establish a foundation for future quantitative and qualitative research in this field and to stimulate applied studies aimed at improving outcomes.

## Study limitations

As this is a study protocol, we have now included a section highlighting possible limitations, including potential bias, loss to follow-up, and limited generalizability. We recognize that these factors may affect the study's outcomes and interpretation, and we will address them carefully in the final study report.

## Acknowledgments

We extend our gratitude to the University of Social Welfare and Rehabilitation Sciences and Lorestan University of Medical Sciences for their invaluable support, which was essential for the completion of this study.

## Author contributions

**Data curation:** Mohammad Saatchi.

**Formal analysis:** Mohammad Saatchi.

**Investigation:** Hamidreza Khankeh.

**Supervision:** Hamidreza Khankeh, Mohammad Saatchi, Mehrdad Farrokhi, Timoor Hosseini.

**Writing – original draft:** Hamidreza Khankeh.

**Writing – review & editing:** Amin Talebi, Hamidreza Khankeh.

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
