## [Decision Letter · Decision Letter 0]

26 Sep 2025

Dear Dr. Khankeh,

Thank you for submitting your manuscript to PLOS ONE. After careful consideration, we feel that it has merit but does not fully meet PLOS ONE’s publication criteria as it currently stands. Therefore, we invite you to submit a revised version of the manuscript that addresses the points raised during the review process.

We look forward to receiving your revised manuscript.

Kind regards,

Mohammad Saadati

Academic Editor

PLOS ONE

Journal Requirements:

4. Please amend the manuscript submission data (via Edit Submission) to include author Hamidreza Khankeh.

5. Please amend your authorship list in your manuscript file to include author Hamid Reza Khankeh.

Reviewers' comments:

Reviewer's Responses to Questions

**Comments to the Author**

1. Does the manuscript provide a valid rationale for the proposed study, with clearly identified and justified research questions?

Reviewer #1: Yes

Reviewer #2: Yes

2. Is the protocol technically sound and planned in a manner that will lead to a meaningful outcome and allow testing the stated hypotheses?

Reviewer #1: Yes

Reviewer #2: Yes

3. Is the methodology feasible and described in sufficient detail to allow the work to be replicable?

Reviewer #1: Yes

Reviewer #2: Yes

4. Have the authors described where all data underlying the findings will be made available when the study is complete?

Reviewer #1: Yes

Reviewer #2: Yes

5. Is the manuscript presented in an intelligible fashion and written in standard English?

Reviewer #1: Yes

Reviewer #2: Yes

You may also provide optional suggestions and comments to authors that they might find helpful in planning their study.

Reviewer #1: Dear authors,

This manuscript addresses an important topic with potential public health and clinical implications especially in the context of Iran. However, despite its relevance, the manuscript has several methodological, structural, and clarity-related issues that must be addressed:

1. The current title is too lengthy and complex. Consider simplifying it to something like: "30-Day Survival and Injury Severity in Traffic Accidents in Iran: A Mixed-Method Study Protocol".

2. The manuscript inconsistently uses both '2025' and '1404'. It’s recommended to clarify this at the beginning by stating 1404 is equivalent to 2025, and then stick to one format throughout the text.

3. Ethical approval details are mentioned but not specific (e.g., reference number).

4. The qualitative analysis section does not clearly explain coding, category development, or how trustworthiness will be established. Add detail on data analysis steps, such as coding strategies and verification methods (e.g., member checking, triangulation).

5. The Cox model design omits discussion of potential interaction terms (e.g., age × ISS). Consider planning for interaction effect testing in the regression analysis.

6. Even as a protocol, the study should acknowledge possible limitations such as potential bias, loss to follow-up, or limited generalizability

7. If data collection tools such as the researcher-made checklist are central to the study, include an English version as an appendix or describe it in more detail.

8. While most of the manuscript is formal, some parts are explanatory or informal. A thorough language revision is needed to ensure consistency.

Reviewer #2: Thank you for your kind opportunity to review the manuscript titled "Investigating 30-day survival rate and its relationship with injury severity in traffic accident victims in Khoramabad in 2025 and Challenges and Applied Recommendations: A mixed method study protocol".

Overall, this is a clear, concise, and well-written protocol. The introduction is relevant. Sufficient information about the previous study findings is presented for readers to follow the present study rationale and procedures. The methods are appropriate details. Article have a clear writing style with sufficient headers and sub- headers such that the reader can readily follow the flow. The description of the methods includes sufficient detail about the procedures used and present these in a logical order. Preregistration of Research Protocols can limit publication bias.

I have some few comments to the authors which may help to improve the manuscript.

Please:

1-provide reference for Sample size.

2-Details of the aims can be the same in everywhere of the protocol.

3-Provide diagram of the data collection and study steps as a figure.

Best regards,

**Do you want your identity to be public for this peer review?** For information about this choice, including consent withdrawal, please see our Privacy Policy

Reviewer #1: No

Reviewer #2: No

---

## [Author Response · Author response to Decision Letter 1]

17 Oct 2025

Thank you for the opportunity to revise our manuscript. We have addressed all comments from the reviewers and editor in detail in the attached “Response to Reviewers” document.

---

## [Decision Letter · Decision Letter 1]

28 Nov 2025

30-Day Survival and Injury Severity in Traffic Accidents in Iran: A Mixed Methods Study Protocol

PONE-D-25-30550R1

Dear Dr. Khankeh,

We’re pleased to inform you that your manuscript has been judged scientifically suitable for publication and will be formally accepted for publication once it meets all outstanding technical requirements.

Kind regards,

Mohammad Saadati

Academic Editor

PLOS ONE

Additional Editor Comments (optional):

Reviewers' comments:

Reviewer's Responses to Questions

**Comments to the Author**

1. Does the manuscript provide a valid rationale for the proposed study, with clearly identified and justified research questions?

Reviewer #2: Yes

2. Is the protocol technically sound and planned in a manner that will lead to a meaningful outcome and allow testing the stated hypotheses?

Reviewer #2: Yes

3. Is the methodology feasible and described in sufficient detail to allow the work to be replicable?

Reviewer #2: Yes

4. Have the authors described where all data underlying the findings will be made available when the study is complete?

Reviewer #2: Yes

5. Is the manuscript presented in an intelligible fashion and written in standard English?

Reviewer #2: Yes

You may also provide optional suggestions and comments to authors that they might find helpful in planning their study.

Reviewer #2: Hello,

I would like to extend my gratitude to the authors for their efforts in addressing the previously mentioned points. The revisions have been thoughtfully made, and the improvements are evident.

Best regards

**Do you want your identity to be public for this peer review?** For information about this choice, including consent withdrawal, please see our Privacy Policy

Reviewer #2: No

---

## [Editor Report · Acceptance letter]

PONE-D-25-30550R1

PLOS One

Dear Dr. Khankeh,

I'm pleased to inform you that your manuscript has been deemed suitable for publication in PLOS One. Congratulations! Your manuscript is now being handed over to our production team.

Kind regards,

on behalf of

Dr. Mohammad Saadati

Academic Editor

PLOS One